# Accumulation of Phenolic Compounds and Antioxidant Capacity during Berry Development in Black ‘Isabel’ Grape (*Vitis vinifera* L. x *Vitis labrusca* L.)

**DOI:** 10.3390/molecules25173845

**Published:** 2020-08-24

**Authors:** Aynur Kurt-Celebi, Nesrin Colak, Sema Hayirlioglu-Ayaz, Sanja Kostadinović Veličkovska, Fidanka Ilieva, Tuba Esatbeyoglu, Faik Ahmet Ayaz

**Affiliations:** 1Graduate School of Natural and Applied Sciences, Biology Graduate Program, Karadeniz Technical University, 61080 Trabzon, Turkey; aynur-leyla@hotmail.com; 2Department of Biology, Karadeniz Technical University, 61080 Trabzon, Turkey; ncolak@ktu.edu.tr (N.C.); sha@ktu.edu.tr (S.H.-A.); faa@ktu.edu.tr (F.A.A.); 3Faculty of Agriculture, University “Goce Delčev”-Štip, 2000 Shtip, Republic of North Macedonia; sanja.kostadinovik@ugd.edu.mk (S.K.V.); fidanka.ilieva@ugd.edu.mk (F.I.); 4Institute of Food Science and Human Nutrition, Gottfried Wilhelm Leibniz University Hannover, 30167 Hannover, Germany

**Keywords:** grape, anthocyanins, proanthocyanidins, seed, skin, development, *Vitis vinifera*, *Vitis labrusca*, Black Sea, antioxidant activity

## Abstract

Grapes are one of the most economically important fruits in the world and are of considerable benefit to human health due to their phenolic compounds. The black ‘Isabel’ grape (*V. vinifera* L. *x*
*V. labrusca* L.) is widely grown in the Black Sea region of Turkey, where it is attracting increasing interest. The aim of this study was to investigate phenolic compounds and antioxidant capacity (DPPH (2,2-diphenyl-1-picrylhydrazyl) and CUPRAC (CUPric Reducing Antioxidant Capacity)) in the grape during berry development, which has been not reported yet from the region. A trend towards an increase in total phenolic compounds, flavonoid, and anthocyanin contents and antioxidant capacity values was observed from un-ripened to overly ripened berries. In addition, anthocyanins in berries and proanthocyanidins seeds were characterized during berry development. Accordingly, malvidin-3-*O*-glucoside was the major anthocyanin in skin (1.05–1729 mg/kg fresh weight (fw)) and whole berry (0.23–895 mg/kg fw), followed by the content of peonidin-3-*O*-glucoside (0.98–799 mg/kg fw and 0.15–202 mg/kg fw, respectively). After veraison (onset of ripening or change of color in the grape berries), all proanthocyanidins showed a gradual decrease through ripening. The results showed that anthocyanins and proanthocyanidins in the grape showed significant stage-dependent changes with positive or negative strong correlations. Considering the phenolic compounds, an optimum harvest date of the grape might be suggested between mid-September and mid-October (263 and 293 DAFBs).

## 1. Introduction

Grapes (*Vitis* spp L.) are one of the most important horticultural crops across the world and have widely received considerable attention because of their polyphenols [1,2,3]. Polyphenols are a major source of phytochemicals in grapes, mainly stilbenes (resveratrol), flavanols, proanthocyanidins, anthocyanins, and phenolic acids in varying amounts in the skin, flesh, and seed [3]. They show a wide spectrum of health promoting properties, including antioxidant, anti-inflammatory, anticarcinogenic, and antibacterial activities [1,2,3,4]. The annual world grape production for 2017 exceeded 77 million tons. Turkey is one of the world’s three largest grape producers, at nine million tons [5].

The *Vitis* genus is represented by 80 species, composed of two sub-genera, *Muscadinia* Planch. and *Euvitis* Planch. Most cultivated grapevines belong to the *Euvitis* sub-genus, including two particularly economically important cultivated grapevines, *V. vinifera* (Eurasian group), which accounts for most of the world’s *Vitis* varieties, and *V. labrusca* (American group). However, species belonging to a third group (East Asian group) are of limited importance in viticulture [5]. It has been reported that *V. labrusca* including its hybrids and *V. vinifera* grapes (Isabel, Concord, Bordô, Niagara, etc.) are the main cultivars used in the grape processing industry (for use in wine, juice, raisins, table grapes, grappa, etc.) in the world’s 10 largest producers (more than 85% in the USA and Brazil) [2].

The black ‘Isabel’ or locally known ‘Isabella’ grape (*V. vinifera × V. labrusca*) is widely cultivated in the Black Sea region of Turkey. It is consumed as a table grape, marmalade, molasse, pickle, jam or juice, depending on local needs. The berry of ‘Isabel’ grape has a distinct aroma, a foxy flavor, a thick and slip skin with a dark or purplish black color, and a sweeter flesh. These particular characteristics of the berry have prompted the inhabitants of the region to extend its cultivation, benefiting from its high adaptability in primarily humid local growing conditions [6]. Also, it has a market value and is therefore widely sold as a table grape in coastal cities in the region.

Grape berries are important in terms of their polyphenols, and differences in phenolic contents during berry development which affected wine or juice quality [3]. In addition, factors including development or ripening, irrigation, soil type, and topography exert positive or negative effects on the content and composition of grape polyphenols and have an important impact on the evaluation of grape varieties and/or clones of a particular variety. Many studies have evaluated the effects of these factors on phenolic composition in wine and table grapes [1,2,3,4,7]. Phenolic compositional changes occurring in grapes during berry development are attracting increasing interest worldwide owing to their economic importance to wine, juice, and table grapes [3,8]. However, few or no data are available concerning the phenolic composition of similar or different grape varieties widely grown in different regions of the world.

Several studies have been reported on the anthocyanin or non-anthocyanin compounds (flavonols, flavan-3-ols, proanthocyanidins, stilbenes, etc.) of ‘Isabel’ grapes (*V. labrusca* or its *V. vinifera* hybrids) from other regions in the world [2,9,10]. To the best of our knowledge, no previous studies have been performed to identify the phenolic constituents of the present hybrid ‘Isabel’ grape (*V. vinifera* x *V. labrusca*) from the Black Sea region (Turkey). The purpose of the present study was therefore to profile the anthocyanin and proanthocyanidin composition of different grape berry parts (skin, whole berry, and seed) during berry development.

## 2. Results and Discussion

### 2.1. Phenolic Compounds Content and Antioxidant Capacity during Berry Development

The changes and variations in total phenolic compounds (TPC), total flavonoid (TF) and anthocyanin (ACY) contents in the whole berry, skin and seed increased gradually and significantly (*P* < 0.05) during berry development (161–293 DAFBs) (Table 1). Their contents were relatively low in the unripe berries (161–201 DAFBs), but subsequently leveled to their maximum values in the ripe and over-ripe berries (293 DAFB). Variety is one of the major factors affecting phenolic compound contents in grapes [4,7]. Red varieties of *V. vinifera* and *V. labrusca* grapes have been reported to have a much higher phenolic content (avg. ~82 mg gallic acid equivalents (GAE)/100 g fresh weight (fw)) than white varieties (avg. ~34 mg GAE/100 g fw) [11]. In agreement with these reports, we found higher TPC content in the present grape berry than that previously reported for several *V. labrusca* varieties (avg. 61.2 mg GAE/100 g fw) including red ‘Isabel’ (57 mg GAE/100 g fw) [11].

The total TF content in the grape was also affected by berry development. Higher TF contents were particularly determined after 232 DAF, representing under-ripe berries, reaching maximum values in the over-ripe berries at 293 DAFB in the whole berry (32 mg quercetin equivalents (QE)/100 g fw) and skin (93 mg QE/100 g fw). The present grape berry had a relatively significantly (*P* < 0.05) high total TF content in the seed in the ripe and over-ripe berries (117 and 118 mg QE/100 g fw, respectively), but the content was insignificant in comparison to unripe and under-ripe berries (Table 1). 

The total ACY contents in the present grape berry (black ‘Isabel’) are also summarized in Table 1. The ACY contents of the grape berry significantly (*P* < 0.05) gradually increased during berry ripening, in particular remaining relatively high in the over-ripe berries in the whole berry (42 mg cy-3-glu equivalents/100 g fw) and skin (171 mg cy-3-glu equivalents/100 g fw) in comparison to unripe berries (Table 1). Pigment color changes in grape berries during development are closely associated with increasing anthocyanin content and affect the overall quality of wine or juice [3,12,13,14,15,16]. The total ACY contents in the present grape berry (black ‘Isabel’) are within the ranges reported for other *V. vinifera* and *V. labrusca* cultivars [11,17,18]. Total ACY content in the present grapes were relatively lower than that reported by Nile et al. [18] for ‘Isabel’ grape from Korea. However, Burin et al. [11] observed rather lower ACY content in several *V. labrusca* varieties.

As shown in Table 1, the antioxidant capacity of the phenolic compounds was also significantly affected by the berry development and ripening. Likewise, antioxidant capacity values also exhibited an identical trend to that of total phenolic contents. In particular, due to their flavonoids, the seeds exhibited a much higher antioxidant capacity than the phenolic extract of the whole berry of the black ‘Isabel’ grape. Since the berries at the first two unripe stages (161 and 181 DAFBs) did not contain any well-developed seed, there were no measurable levels of TPC and TF contents or antioxidant capacity value at those stages. The phenolic contents (TPC, TF, and ACY) of the whole berry and the skin exhibited strong linear relationships (range; *R^2^* = 0.848–0.991) and were positively significantly strong correlated (range; *r* = 0.921–0.987, *P* < 0.01, 0.05) with antioxidant capacity values assayed by DPPH (2,2-diphenyl-1-picrylhydrazyl) and CUPRAC (CUPric Reducing Antioxidant Capacity). Similarly, the TPC and TF contents exhibited high and strong linear relationships (range; *R^2^* = 0.624–0.934) and were significantly correlated with the antioxidant capacity values (range; *r* = 0.790–0.967, *P* < 0.01, 0.05) (Table 1). In agreement, Tomaz et al. [3] reported that the higher the phenolic content of a fruit, the greater its antioxidant capacity.

### 2.2. Anthocyanin Composition during Berry Development

Anthocyanin profile in the skin of the black ‘Isabel’ grape is given in Table 2. Only 3-glucosides of five anthocyanidins usually found in *V. vinifera* and *V. labrusca* varieties (delphinidin, cyanidin, petunidin, peonidin, and malvidin) were detected in the present grape. It has been known that *V. vinifera* grapes are composed of anthocyanidin-3-*O*-monoglycosides as major anthocyanins, whereas diglycosides are mainly present in other grapes, such as *V. amurensis, V. riparia, V. rupestris*, and their hybrids. Furthermore, the color of some hybrids such *V. vinifera* x *V. labrusca* is dependent mainly of monoglycosides since diglycosides are more susceptible for browning [19,20]. A general trend toward a significant increase in the levels of anthocyanins in the whole berry and skin was confirmed during development. 

The dominated anthocyanin was malvidin-3-*O*-glucoside (mv-3-glc), followed by peonidin-3-*O*-glucoside (pn-3-glc). Concentrations of mv-3-glc ranged from 1499 to 1729 mg/kg fw in the skin and from 560 to 895 mg/kg fw in the whole berry in the over-ripe berries at 263 DAFB (20 Sept.) followed by concentrations of pn-3-glc (596 to 799 and 132 to 202 mg/kg fw, respectively) (Table 2). Overall, the levels of anthocyanins in the berry increased gradually and slightly at six developmental stages (161–293 DAFBs) in particular, being relatively significantly high (*P* < 0.05) in ripe and over-ripe berries (263 and 293 DAFBs). Our findings show that the anthocyanin profile of the black ‘Isabel’ grape resembles those exhibited by the most widespread *V. vinifera* and *V. labrusca* red grape varieties reported by many previous authors [1,2,4,7,10,11,15,16,17,18,19,20,21,22,23,24]. The present findings are also in good agreement with Zhu et al. [7] who observed lower mv-3-glc (avg. 20, range 0-201 mg/kg fw) and pn-3-glc (avg. 78, range 13–135 mg/kg fw) in North American grapes, including *V. labrusca*, than in the present ‘Isabel’ grape. Moreover, the presence of the two major anthocyanins in the present ‘Isabel’ grape was also reported in ‘Isabel’ grapes from Brazil by Lago-Vanzela et al. [2] and Yamamoto et al. [10]. Our results for monoglycosides anthocyanins were in good agreement with those of Wang’s research group [25], who observed the same fluctuation of concentrations of delphinidin and cyanidin derivatives in “Kyoho” grape skin. All these reports have indicated that variation in anthocyanin content or composition in grapes has been attributed to several external factors, including light, temperature, crop size, leaf area, pathogen infection (phylloxera, bacteria, nematodes-mites, etc.) and plant growth regulators that influence the synthesis and accumulation of anthocyanins [2,6,26]. Furthermore, Shoeva et. al [27] demonstrated that even in universal metabolic pathways, some species-specific regulatory features of structural genes are present. Their results indicated that, in the anthocyanin biosynthesis pathway (ABP), genes may be regulated by ABP-specific regulatory factors, and their expression levels may be strongly associated with anthocyanin pigmentation, or they may be expressed independently of pigmentation. It has been known that ‘Isabella’ grape varieties are resistant to fungal diseases and they can be grown in regions with cool and humid climates [28]. ‘Isabel’ (or Isabella) grape varieties used for grape cultivation in Black Sea region are not affected by fungal diseases, there is no need for the use of drugs in the grape’s cultivation, but some nematodes can cause serious leaf and fruit losses, especially in black ‘Isabel’ grape (reference: personal communication with authorities in Republic of Turkey of Ministry of Agriculture and Forestry-Trabzon Directorate of Agricultural Organization).

A detailed study about phenolic contents of European, East-Asian, Euro-Asian, North American, Euro-American and Muscadine grapes has revealed presence of 45 anthocyanin compounds including monoglucosides, diglucosides and their acylated derivatives of six anthocyanidins [7]. Accordingly, mv-3-glc and mv-3,5-diglc were the major anthocyanin types skins of *V. vinifera* (77.8%) and East-Asian species (53.55%), respectively. The study also reported that the major anthocyanin types in most North American grapes were cyanidin (43.21%) and delphinidin derivatives (32.20%), cy-3-glc and dp-3-glc as the main anthocyanins. In the case of *V. rotundifolia* grapes, only the diglucosides of the six anthocyanidins were found, where dp-3,5-glc was the most abundant anthocyanin in ‘Alachua’ (45.16%) and cy-3,5-glc was the most abundant in ‘Noble’ (36.17%) followed by pt-3,5-glc (avg. 21.44%), respectively. In the skins of the hybrids, proportions of mv-, dp- and pt-derivatives, were also different in Euro-Asian hybrids (24.20%, 43.34% and 19.59% of total anthocyanins, respectively) and Euro-American hybrids (47.56%, 24.54% and 21.24% of TAs, respectively). With regard to monoglucosides and diglucosides anthocyanins, there was a distinct separation among the different species. The *V. vinifera* grapes only contained the monoglucoside anthocyanins, whereas *V. rotundifolia* grapes only diglucosides anthocyanins. Among the East Asian species and *V. amurensis* × *V. vinifera* hybrids, diglucosides anthocyanins were dominant in most grapes (accounting for 86.81% of TAs on average), except *V. quinquangularis* ‘Mao’ and the hybrids ‘Zuohongyi’ and ‘Hasang’. While monoglucoside anthocyanins were the main anthocyanins in North American grapes (*V. labrusca*) and Euro-American hybrids (accounting for 74.76% of TAs on average), except *V. aestivalis* ‘Black Spanish’ and the hybrid ‘St. Croix’ [7]. As indicated, ‘Isabel’ is a hybrid grape cultivar (*V. vinifera* × *V. labrusca*) and the occurrence of anthocyanidin 3,5-diglucosides is expected, in line with previous findings [9] and other hybrid grape cultivars [2,27,29]. Due to that, the quantification of these compounds is made using the calibration curve of the most similar compound: malvidin 3-glucoside is used for anthocyanidin 3-glucosides, and malvidin 3,5-diglucoside for anthocyanidin 3,5-diglucosides. However, a previous study [10] showed that anthocyanin monoglucosides were the main anthocyanins of ‘Isabel’ grape juice, with a lower proportion of anthocyanin- diglucosides. Our findings, together with this literature, explain why the present grape (Isabel) does not have diglucoside anthocyanins.

### 2.3. Seed Proanthocyanidins during Berry Development

Proanthocyanidins (PAs), such as monomers (catechin, epicatechin, gallocatechin, epigallocatechin, epigallocatechingallate), dimeric procyanidins (B1, B2, B5, B7) galloylated dimeric procyanidins, trimeric procyanidins, galloylated trimeric procyanidins, tetrameric to hexameric procyanidins as well as gallic acid, caffeic acid and quercetin were detected in the ‘Isabel’ grape seeds, investigated in the present study, using LC-MS/MS and identified as described by Cádiz-Gurrea et al. [30] and Pantelić et al. [31] (Table 2). The corresponding HPLC-chromatograms of the seed polyphenols at different development stages of ‘Isabel’ grape at 280 nm with the main proanthocyanidins are shown in Figure 1 and Figure 2.

Overall, during berry development, the concentrations of flavan-3-ols were in an increasing trend and polymeric compounds were in a decreasing trend. Flavan-3-ols in seeds increased gradually up to the final development stage of unripe berries (161–201 DAFBs, range; 3.1% to 13%), after which the concentration remained low, ranging from 4.39% to 3.06% in the ripe and over-ripe berries (232–293 DAFB). On the contrary, the polymeric proanthocyanidins gradually declined through the development (161–232 DAFBs, 83%–19%) This phenomenon can be easily seen on a Diol column (Figure 2). The concentration of non-polymeric polyphenols decreased slightly or else remained at more or less at the same concentrations through the unripe and the first ripe development stages (Table 2). Their concentrations then increased suddenly at 201 DAFB, following the same pattern with fluctuations. The relationship between seed growth and berry development is reported to vary within and among varieties, and may also indicate differences in seed polyphenols [12]. Considering the gradual linear increases and decreases in anthocyanin concentrations, the seed polyphenols did not exhibit such a linear pattern during maturation and ripening of the present grape (Isabel). The fluctuations may be due to significant differences in metabolism, composition, berry density and size, cluster position, and environmental conditions [12].

### 2.4. Principal Component Analysis (PCA) of the Anthocyanins and Proanthocyanidins during Berry Development

To identify and characterize the possible associations and correlations between berry development and anthocyanins (Figure 3A) or proanthocyanidins (Figure 3B) profiles, a principal component analysis (PCA) was performed. Factor analysis (F) was performed on the basis of the matrix of correlation coefficients using principal component factor (Table 3). The matrix of dominant rotated factor loadings of anthocyanins (variables) and stage of berry development (observations) are also shown in Table 4. Only one factor was identified for anthocyanins explained 96.76% of the total variance of the original data (see Table 4). Two factors were identified for berry development. Factor 1 formed the biggest association composed of berry development, except underripe stage (232 DAFB) which belong to Factor 2 (0.983), which explained again 96.76% of the variance of the data (Table 4). The PCA and the correlation matrix, which explained 99.30% of total variance where PC1 accounts for 96.76% of the variance and PC2 for 2.54%, confirmed that the anthocyanins were closely associated and positively significantly strong correlated (range; *r* = 0.890–0.999, *P* < 0.05) with the ripe and overripe grape berries (263 and 293 DAFBs). In particular, the major two anthocyanins, mv-3-glc and pn-3-glc, in the skin and whole berry were dispersed at the right upper plan on PC1 and the remaining other six anthocyanins were at the right lower plan on PC1. In the case of other four berry development stages, they were dispersed at the left upper (unripe, 161–201 DAFBs) and lower (underripe 232 DAFB) plans on PC2 with negative loadings and no association (Figure 3A).

The PCA loading plot of the seed proanthocyanidins (PA) contents during the six stages of berry development are presented in Figure 3B. The PCA explained 85.33% of the variance; specifically, PC1 and PC2 accounted for 49.24% and 36.02% of total variance, respectively. Accordingly, seven PAs at the right upper plan with the last unripe berry and two PAs at the right lower plan with the underripe, ripe and overripe berry developments with positive loadings on PC1 were closely associated and correlated, while remaining six PAs at the left upper plan was closely associated and correlated with only first two unripe berry developments with negative loadings on PC2 (Figure 3B). Only dimer B5, at the left lower plan on PC2 with negative loadings, did not associate and correlate with any berry development.

The matrix of correlation coefficients and the data of principal component factor analysis, as well factors of berry development stages of the PAs are shown in Table 5. As it can be seen from the correlation coefficients values, there were some significant positive (range; *r* = 0.821–0.980, *P* < 0.05) and negative (range; *r* = –0.829 to –0.918, *P* < 0.05) correlations of the PAs in association with the aforementioned berry developments. Through the correlation matrix of the data it was possible to observe that there is a representative number of values higher than 0.500, which means a correlation between the variables ranging from moderate to strong. The contribution of each variable in the PCA was evaluated through the factor (F) analysis (Table 6). Accordingly, four factors were identified for proanthocyanidins: Factor 1 with dimer B2, dimer B5, and tetramers A2 and Factor 2 with trimers and tetramers formed the biggest association, while the Factor 3 and 4 had the lowest association (Table 6). Three factors were identified for berry development: Factor 1 with unripe (161 DAFB), F2 with ripe and overripe (263 and 293 DAFBs) berry development formed the biggest association, while it was the lowest for Factor 3 (Table 6).

Grape berry development can be divided into three growth stages; an increase in pericarp and seed cell numbers (stage I), the development of the seed embryo and hardening of the seed coat (stage II) and the ripening stage (stage III) characterized by rapid accumulation of sugars. The inception of stage III, known as veraison (onset of ripening or change of color in the grape berries), is also characterized by berry softening in parallel with anthocyanin accumulation (red varieties). At this stage most of the seeds are fully developed. Rapid accumulation of anthocyanins has been reported in veraison [3,4,5,6,7,8,9,10,11,12,32].

In the present grape (Isabel), veraison occurred at 232 DAFB (20 August 2018), when the berries were visually characterized by a sudden change in skin color (full red), whitish-reddish soft flesh, and developed seeds [6]. This is in good agreement with findings reported by Katalinić and Petar [32] for ‘Plavac Mali’ and ‘Trnjak’ observed at veraison (15–26 August) in the same date range for the present grape berry (Isabel). Katalinic and Petar [32] noted a marked decrease in flavonols in the seeds of the two grape cultivars after veraison in parallel with relatively high concentrations of monomeric forms in the grape seeds that dropped rapidly after veraison and remained relatively constant during maturation. Other authors have also concluded that seed maturation is characterized by relative enrichment in dimeric compared to monomeric forms [1,2,3]. However, in some cases, polyphenols in grape seeds do not exhibit a similar trend of changes during development [1]. For example, flavan-3-ols and procyanidins exhibit a gradual decrease during ripening of ‘Cabernet Sauvignon’ grape berry seed immediately after the inception of the grape berry to veraison (17–19 August 1998) [12]. In general, the total content of proanthocyanidins found in *V. labrusca* varieties (e.g. Bordô grapes) seems to be lower than the usual values reported for *V. vinifera* grape cultivars [2]. In ‘Isabel’ grape from Brazil, Yamamoto et al. [10] reported that the major procyanidin was dimer B4 (range; 4.18–3.78%), followed by B2 (range; 0.93–1.17%) and B1 (range; 0.78–0.91%) through two vintages. However, in the present grape (black ‘Isabel’), the contents of both dimers B1 and B2 were much lower than the reported, and B4 was not detected. Unlike the report, in the seeds of black ‘Isabel’ grape, dimers B5 and B7 were detected at high levels. In the present grape, overall, the mass amounts of monomeric and dimeric flavan-3-ols depend upon variety, vintage, soil type, climate parameters, regional factors, topography, and degree of maturity, with some of these factors exhibiting a positive influence and others a negative one. It is therefore essential to determine their effects through chemical analysis [1,2,3,4,7].

Endogenous sugar increases the synthesis of anthocyanin, a phenomenon directly and closely related to peonidin derivatives and the other phenolic compounds present in grape skin [26]. Anthocyanin concentrations increase during ripening until reaching a maximum value, after which the concentration decreases, particularly in over-ripe grapes [33]. The increasing trend in anthocyanin content after veraison (15–26 August) observed in the present study was similar to that in the sugar increase and acid decrease in the same grape variety (‘Isabel’, black) picked from the same grapevine the same number of days after full bloom (DAFB). Those data and the present results confirmed an inverse correlation between decreased acid and sugar or anthocyanin content [6,26].

Proanthocyanidins are responsible for the astringent and bitter properties of wine and are released from grape skins and seeds [1,2,26,34]. These are mainly (+)-catechin (C), (−)-epicatechin (EC), (−)-epicatechin-3-*O*-gallate (ECG) and (−)-epigallocatechin (EGC) linked by C(4)→C(6) or C(4)→C(8) interflavanoid bonds. Lower contents are reported in skins than in seeds. It has been indicated that both C and EC are the major flavan-3-ols in *V. labrusca* red grapes [2,10]. Our results were in agreement with those reported by the authors. These seed tannins are usually composed of C, EC, and ECG, while skin tannins are generally larger with a higher mean of polymerization (mDP). EC is the major extension subunit in skins, while seeds have been found to contain similar amounts of C and EC subunits [34]. Berry development has been associated with the polymerization of these particular phenolic compounds, which lead to a marked decrease in astringency [33]. It has been reported that biosynthesis of tannins occurs after anthesis, reaching a maximum at veraison [34]. In ‘Shiraz’ grape seeds, the highest concentration of flavan-3-ols was observed at one week post-veraison, with a subsequent decline until maturity, while in skins, the highest concentration was observed before veraison followed by a continuous decrease until complete ripeness [34]. In agreement with these findings, the total proanthocyanidin content with polymers (TPAs + pol) or without polymers (TPAs – pol) in seeds of the ‘Isabel’ grape in the present study (see Table 2 and Figure 3B) decreased gradually as the berry entered ripe development just after veraison (after 232 DAFB). These results and reports have confirmed that accumulation of phenolic compounds in grapes exhibit considerable variation that might be dependent on some factors such as canopy management, irrigation, soil type (in particular pH), climate conditions, age of grape vine, berry development, and pathogen effects have a profound impact on the content and compositions of phenolic compounds in grape berries [3,33,34].

## 3. Materials and Methods

### 3.1. Chemicals

The following chemicals were used in this study: Acetonitrile CHROMASOLV^™^ gradient grade ≥ 99.9% (Honeywell Riedel-de Haën^™^, Seelze, Germany), acetic acid Optima^®^ LC/MS glacial (Fisher Chemical, Geel, Belgium), acetone Reag. Ph. Eur. 100% (VWR, Fontenay-sous-Bois Cedex, France), methanol ≥99.8% HPLC grade (Fischer Scientific, Loughborough, United Kingdom), water (deionized, nanopure^®^, Werner, Leverkusen, Germany).

(−)-Epicatechin was purchased from Sigma (Steinheim, Germany). (+)-Catechin, procyanidin dimer B1, procyanidin trimer C1, gallic acid, quercetin, cyanidin-3-glucoside and cinnamtannin A2 were obtained from PhytoLab GmbH & Co. KG (Vestenbergsgreuth, Germany). Dimer B2, B5 and B7 were kindly provided from Technische Universität Braunschweig (Braunschweig, Germany). The anthocyanin reference compounds delphinidin-3-*O*-glucoside, cyanidin-3-*O*-glucoside, petundin-3-*O*-glucoside, peonidin-3-*O*-glucoside and malvidin-3-*O*-glucoside were purchased from Phytolab (Vestenbergsgreuth, Germany). All these pure standards were used for identification.

### 3.2. Plant Material

Berries of the black ‘Isabel’ grape (*V. vinifera* L. x *V. labrusca* L.) were sampled from 20-year-old grapevines in 11 locations (Ordu, Giresun, Trabzon, Rize and Artvin, at 151–220 m a.s.l) as previously described in our recent study [6], from northeast and northwest Anatolia in Turkey. Based on days after full bloom (DAFB) analysis, six fruit development stages were investigated as previously described [6], divided into four distinct specific categories (unripe; 161–201 DAFBs (skin green), under-ripe; 232 DAFB, ripe; 263 DAFB (reddish) and over-ripe; 293 DAFB (dark blue black) (Figure 4)), in which the grape berries were harvested at 20-day intervals for unripe berries followed by 30-day intervals thereafter (201–293 DAFBs). Randomly selected bunches of moderate size and weight (2–3 kg) were handpicked in triplicate from the study locations, and were then combined and harmonized for each stage. The berries were immediately washed free of any residues (dead flower debris or decayed or abnormally developed berries not at the correct stage of development) using distilled water. They were than kept cold (below ∼4 °C) and transported to the laboratory within approximately 2.5–3 hrs. These were first treated with liquid nitrogen (–196 °C), lyophilized (Christ, alpha 2–4 LDplus, Osterode, Germany), and then stored at –80 °C until further analysis.

### 3.3. Extraction of Whole Berries for Anthocyanin Analysis by HPLC-PDA and LC-MS/MS

The method described by Lago-Vanzela et al. [2] with slight modifications was used in the extraction of whole berries. These were first finger-pressed to separate the seeds. The de-seeded separated berries were then immediately homogenized with 3 × 150 mL of the solvent mixture methanol:water:formic acid (50:48.5:1.5, *v*/*v*), followed by 40 min of shaking in the absence of light at room temperature. The whole berry extract was then centrifuged (Hermle Z 326, Wehingen, Germany) at 10,000× *g* at 5 °C for 25 min. The supernatant was dried separately in a rotary evaporator (~30 °C) to eliminate excess methanol, and its volume was brought to 100 mL using deionized water. In order to remove sugars and other polar non-phenolic compounds present in the flesh extract, 5 mL of extract was first diluted with 5 mL of 0.1 N HCl. The prepared sample was then passed through C18 SPE-cartridges (Supelclean™ PSA SPE Bulk Packing, Merck, Germany) that had been previously conditioned with 5 mL of methanol and 5 mL of water. The sample was washed with 5 mL of 0.1 N HCl and 5 mL of water and next eluted with 3 × 5 mL of methanol. The eluate was dried in a rotary evaporator (~30 °C), resolved in 3 mL of 20% methanol in water, and finally directly injected onto the HPLC equipment for anthocyanin analysis.

### 3.4. Extraction of Skin Anthocyanins for Analysis by HPLC-PDA and LC-MS/MS

The method used in the extraction of skin anthocyanins was adopted from Lago-Vanzela et al. [2] with slight modifications. Healthy fresh berries (at least 100 berries) were manually and carefully peeled by hand. The skin was immediately frozen at –80 °C for 12 h and then freeze-dried for 24 h (Christ, alpha 2-4LD plus, Germany). The dried skin samples were homogenized in a porcelain mortar with the aid of a pestle and immersed in 170 mL of a solvent mixture (methanol:water:formic acid, 50:48.5:1.5, *v*/*v*) under an ultrasonic bath for 15 min in a cold environment lower than 8 °C. The samples were then centrifuged at 7000 rpm 5 °C for 15 min, followed by a second extraction of the resulting pellets using the same volume of the extraction solvent. All supernatants were combined, concentrated using a rotary evaporator (Heidolph, Schwabach, Germany), and freeze-dried (Christ, alpha 2-4LD plus, Germany). The dried sample was then diluted with 50 mL H_2_O deionized water and stored at –20 °C for further experimental use. Solid-phase extraction (SPE) of the phenolic extracts obtained was carried out as described by Rodriguez-Soana and Wrolstad [35] with slight modifications using Bound Elut C-18 columns (Agilent, 500 mg, 3 mL, USA) to eliminate sugars, with other polar substances being eluted with water and polyphenolic compounds with ethyl acetate. The anthocyanins remaining in the column were eluted with 0.1 N HCI (1:10, v/v), filtered (0.20 μm, RC-membrane, Ministart RC 4, Sartorius, Germany), and directly injected onto the HPLC column to determine the anthocyanins.

### 3.5. Preparation of Seeds for Proanthocyanidin Analysis by HPLC-PDA and LC-MS/MS

The seeds of a third sample of 100 grape berries were separated, washed with water, and dried softly with paper. Before analysis, all grape seeds were treated with liquid nitrogen and freeze-dried. The freeze-dried samples were again treated with liquid nitrogen and ground with a laboratory mill (IKA A 11 basic, Staufen, Germany) for 30 sec. The extraction of proanthocyanidins from the seed samples were performed according to Curko et al. [36] with slight modifications. Approximately 0.5 g of the powdered seeds was extracted with 3 × 10 mL of 70% (*v*/*v*) aqueous acetone for 20 min with an ultrasonic homogenizator (Branson Ultrasonics Sonifier 450, Danbury, USA) and centrifuged for 5 min at 8000 rpm. A second extraction was done with the residue. The residues were combined, evaporated and freeze-dried. The samples were stored at –20 °C until analysis.

### 3.6. Analysis of Anthocyanins in Grape Berries by HPLC-PDA Analysis

The identification and quantification of anthocyanins were performed according to Lago-Vanzela et al. [2]. The analysis was carried out on a Luna 5 μ C18 column (250 × 4.6 mm) with a guard column. A binary gradient of a mixture of water/acetonitrile/formic acid was used. The linear gradient was A 83/7/10; B: 40/50/10 (*v*/*v*/*v*); A%: 0 min. 94%, 20 min. 80%, 35 min. 60%, 35 min. 40%, 40 min. 90%, and 45 min and 55 min 94%. The flow rate was set at 0.5 mL/min. Chromatographic separation and detection were performed using an Agilent Series 1100 chromatographic system equipped with a quaternary pump, a degassing device, a 20-μL injection loop (Rheodyne, Cotati, CA, USA), and a PDA detector (200–650 nm). 

Stock solution of malvidin-3-glucoside (5 mg/L) was prepared with (5% *v*/*v*) perchloric acid solution to construct external standard curve (*y* = 269.37 × –19.353, and correlation coefficient *R^2^* = 0.9997). This solution of malvidin-glucoside was diluted to obtain standard solutions (100–500 µg/mL for the preparation of calibration curves. These solutions were freshly prepared every day. The final anthocyanin concentration is expressed as milligram per kilogram fresh weight (fw) of extracts. All analyses were done in triplicate.

### 3.7. Proanthocyanidin Characterization of Grape Seeds by LC-MS/MS Analysis

Seed samples at a concentration of approximately 10 mg/mL were analyzed with an Agilent system (Waldbronn, Germany) equipped with a G1312A bin pump (Agilent Technologies 1100 Series), a column oven G1316A ColComp (Agilent Technologies 1100 Series), an autosampler G1329B ALS SL (Agilent Technologies 1200 Series) with a thermostat G1330B ALSTherm (Agilent Technologies 1200 Series), and a photo diode array (PDA) detector G1315B PDA (Agilent 1100 Series), coupled with a Bruker HCT Ultra Ion Trap with electrospray ionization. A 5 μL sample was injected onto the Aqua 3 μ C-18, 125 Å, 150 × 2.00 mm 3 μ column (Phenomenex, Aschaffenburg, Germany) protected by a guard column of the same material. The dry gas temperature was set at 325 °C, the drying gas (nitrogen) flow rate was 10.0 L/min, and the nebulizer gas (nitrogen) pressure was 60 psi. In negative mode, the HV capillary voltage was 3000 V, the capillary exit voltage was -21 V, the trap drive was 72.1, the octopole RF voltage was –187.1 Vpp, lens 2 was 60 V, and the HV end plate offset was –500 V. Data were acquired in negative mode (100–2200 *m/z*) and analyzed using Bruker HyStar V3.2 (Bremen, Germany). The mobile phases consisted of (A) water/acetic acid (98/2; *v*/*v*) and (B) acetonitrile. The gradient was 0 min 3% B, 25 min 10% B, 45 min 35% B, 50 min 75% B, 55 min 75% B, 65 min 3% B, and 70 min 3% B with a flow rate of 0.2 mL/min at 25 °C.

### 3.8. HPLC-PDA Analyses of Grape Seed Proanthocyanidins on a Reversed-Phase Column

HPLC analyses were performed on a Jasco system (Gross-Umstadt, Germany) equipped with a PU-2080 Plus Intelligent HPLC Pump, a DG-2080-53 3-Line Degasser, a LG-2080-02 Ternary Gradient Unit, a AS-2057 Plus Intelligent Sampler, a Column-Thermostat Jetstream Plus, and a MD-2010 Plus Multiwavelength Detector. Proanthocyanidins were separated on a reversed phase Aqua 5 µ C-18, 125 Å, 250 × 4.6 mm column (Phenomenex, Aschaffenburg, Germany) at 280 nm protected by a guard column of the same material. The same eluent and gradient were used as in the polyphenol LC-MS/MS analysis, with a flow rate of 0.5 mL/min at 25 °C. Seed samples were diluted in acetonitrile/water (1:9, *v*/*v*; approximately 5.5 mg/mL). Briefly, 20 µl from all samples was injected three times, and (-)-epicatechin standard solutions in the range of 0.5 to 1000 mg/L were injected twice.

### 3.9. HPLC-PDA Analyses of Grape Seed Proanthocyanidins on a Diol-column

The analysis of grape seed extracts was done using a novel HPLC method on a diol stationary phase according to Kuhnert et al. [37]. HPLC analyses were performed on an Agilent system (Waldbronn, Germany) equipped with a G1312A bin pump (Agilent Technologies 1200 Series), a column oven G1316A TCC (Agilent Technologies 1200 Series), an autosampler G1329B 1260 ALS (Agilent Technologies 1260 Infinity) with a thermostat G1330-89011 (Agilent Technologies 1200 Series) and a photo diode array (PDA) detector G1315D DAD (Agilent Technologies 1200 Series). Samples were diluted in methanol/acetonitrile (2:8, *v*/*v*) to a final concentration of approximately 5.5 mg/mL and 10 µl were injected three times. Proanthocyanidins were separated on a normal phase MonoChrom 3 Diol 150 × 2.0 mm (Agilent, Waldbronn, Germany) at 280 nm protected with a guard column of the same material and were eluted with a flow rate of 0.2 mL/min at 25 °C and a gradient mode with the mobile phases (A) acetonitrile/acetic acid (98/2; *v*/*v*) and (B) methanol/water/acetic acid (95/3/2): 0–19 min, 0–16% B, 19–30 min, 16–100% B, 30–40 min, 100% B isocratic, 40–45 min, 100–0% B, 45–50 min, 0% B.

The calibration curve of (+)-catechin was in the range of 4.7–467 mg/l and was used for the quantification of the monomers and galloylated monomers. The calibration curve of procyanidin B1 was in the range of 2.3 to 231 mg/L and was used for the dimeric proanthocyanidins and galloylated dimeric proanthocyanidins. The calibration curve of the trimer procyanidin C1 was in the range of 8.9–298 mg/L and was used for the trimeric proanthocyanidins. The calibration curve of cinnamtannin A2 was in the range of 8.1–272 mg/L and was used for the quantification of tetrameric proanthocyanidins and as an equivalent for pentameric and polymeric proanthocyanidins.

### 3.10. Determination of Phenolic Compounds by Spectroscopic Methods

The TPC content of the grape sample extracts was determined using Folin–Ciocalteu reagent (Sigma, St. Louis, MO, USA) as described by Singleton and Rossi [38] with slight modifications. The phenolic content was calculated as gallic acid equivalents GAE/100 g of fresh material on the basis of a standard curve of gallic acid concentrations ranging from 0.1 to 10 μg mL^-1^ at five points (*y* = 0.0611x + 0.0031, *R*^2^ = 0.9997). The TPC content was expressed as mg GAE/100 g on a fresh-weight basis (fw).

The aluminum chloride colorimetric method was used for the determination of the total flavonoid (TF) content of the sample [39]. The TF content of the same extracts was calculated from a five-point calibration plot using a standard curve of quercetin concentrations ranging from 0.1 to 50 µg mL^−1^
*(y* = 0.0531 × –0.0247, *R*^2^ = 0.9992), and the results were expressed as mg quercetin equivalents (QE)/100 g on a fresh-weight basis (fw).

A pH-differential method was used for determination of total anthocyanin (ACY) content [40]. The anthocyanin content was calculated using the equation anthocyanin content (mg 100 g^−1^) = *A* × MW × DF/(*ϵ* × *W*), where *A* = absorbance (*A*_520 nm_ − *A*_700 nm_)_pH 1.0_ − (*A*_520 nm_ − *A*_700 nm_)_pH 4.5_, MW = molecular weight of cyanidin-3-glucoside (C_15_H_11_O_6_, 449.2 g/mol, cy-3-glc), DF = dilution factor, *ϵ* = molar absorptivity (26,900), and *W* = sample weight (g). The results were expressed as mg cy-3-glc/100 g on a fresh-weight basis (fw).

### 3.11. Determination of the Antioxidant Capacity by DPPH and CUPRAC Assays 

DPPH^•^ (2,2-diphenyl-1-picrylhydrazyl) radical scavenging capacity was determined according to the method described by Blois et al. [41] with some modifications. The DPPH scavenging activity of the sample was calculated as micromole of Trolox equivalents (TE) per gram of fresh weight (µmol TE/g fw) calculated from a five pointed calibration plot of Trolox concentrations ranging from 10 to 320 µM mL^−1^ (Abs*_517 nm_*, y = –0.0019x + 0.624, *R^2^* = 0.9998).

The cupric ion (Cu^2+^) reducing power (CUPRAC, CUPric Reducing Antioxidant Capacity) of the grape berry samples was measured using the method published elsewhere with minor modifications [42]. The results were expressed as micromole of Trolox equivalents (TE) per gram of fresh weight (µmol TE/g fw) calculated from a five-point calibration of Trolox concentrations ranging from 10 to 320 µM mL^−1^ (*y* = 0.0035x – 0.0127, *R*^2^ = 0.9990).

### 3.12. Statistical Analysis

All extractions and analyses were performed in triplicate (*n* = 3). Data were analyzed by analysis of variance using one-way ANOVA and a means comparison with Duncan’s multiple range test (IBM SPSS Statistics Ver. 22.0) at a significance level of *P* < 0.05. A statistical software package was employed to perform principal component analysis (PCA) (XLSTAT statistical and data analysis solution, Addinsoft, 2019, Long Island, NY, USA).

## 4. Conclusions

According to our findings, the phenolics quantified and antioxidant capacity measured during the development of the berry parts exhibited significantly high and strong correlations, linear regressions, and stage-dependent changes. It may be suggested that black ‘Isabel’ grape skin and seed have a high potential antioxidant capacity that is very likely linked to their equally high phenolic compound contents, particularly anthocyanins in the skin. Our data indicate that berry development and ripening around veraison hastened berry coloration, and that the skin and whole berry anthocyanins were dominated by malvidin-3-*O*-glucoside. In addition, during berry maturation and ripening, concentrations of monomeric proanthocyanidins and anthocyanins were in an increasing trend while polymeric compounds were in decreasing trend. This study also suggests an optimum harvest date for the region in which the grape is used as a table grape, or for other purposes, from approximately mid-September to mid-October (between 263 and 293 DAFB) in terms of optimal levels of polyphenols (e.g., anthocyanins). These findings will help horticulturists, food technologists, and nutritionists to compare their results with the present findings for other ‘Isabel’ grape genotypes or *V. labrusca* cultivars from other regions that are rich in these phenolic compounds for future use in improving those cultivars. 

## Figures and Tables

**Figure 1 molecules-25-03845-f001:**
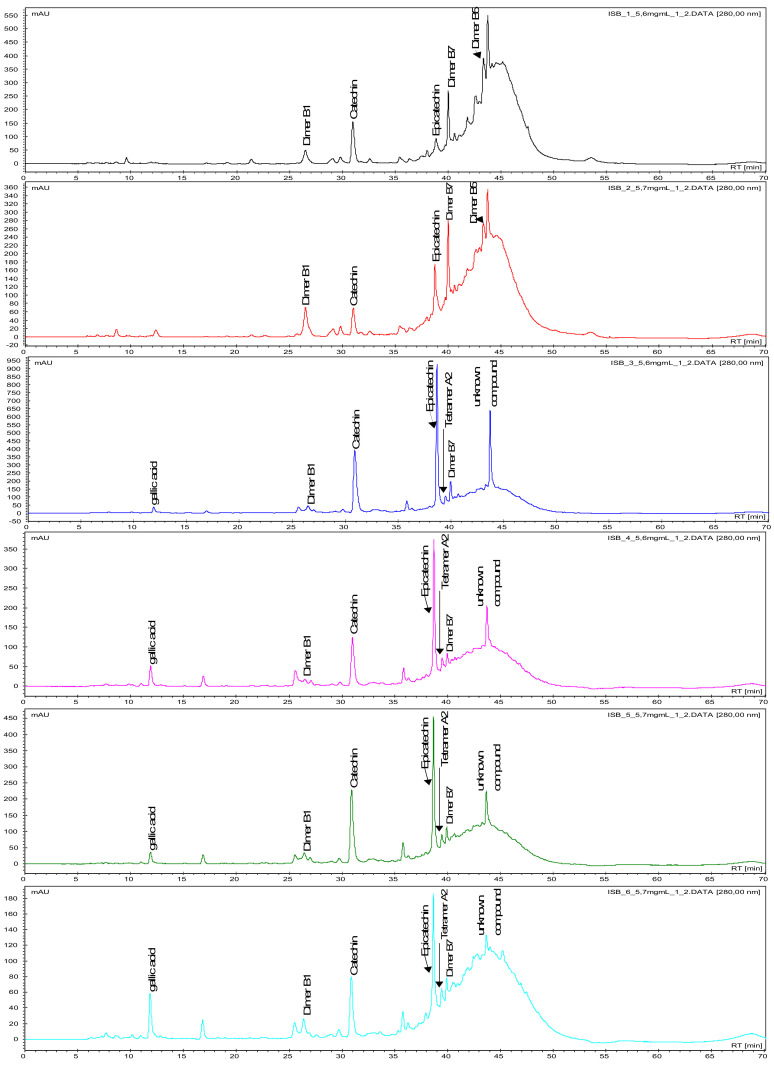
HPLC-chromatograms of proanthocyanidins during berry development and ripening (from top to bottom: six stages 161–293 DAFBs) of black ‘Isabel’ seeds on an Aqua C18-column recorded at 280 nm.

**Figure 2 molecules-25-03845-f002:**
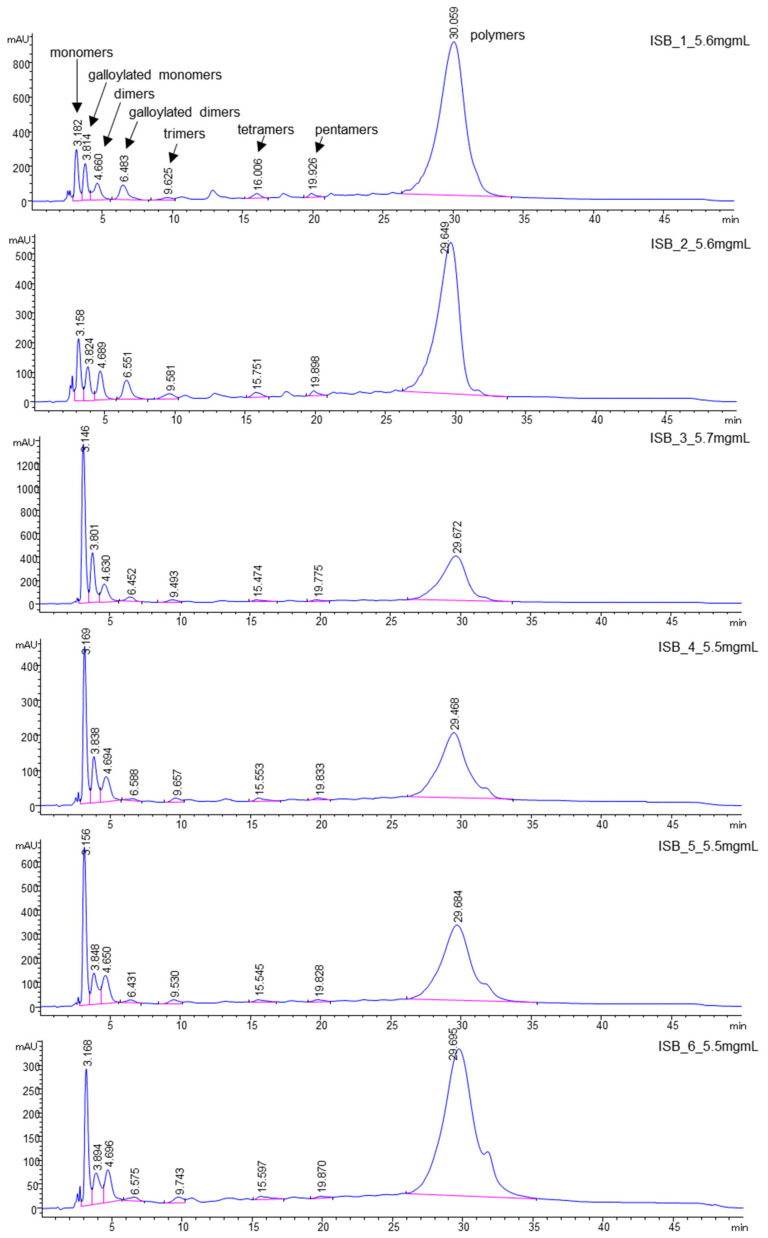
HPLC-chromatograms of proanthocyanidins during berry development and ripening (from top to bottom: six stages 161–293 DAFBs) of black ‘Isabel’ seeds on a Diol column recorded at 280 nm.

**Figure 3 molecules-25-03845-f003:**
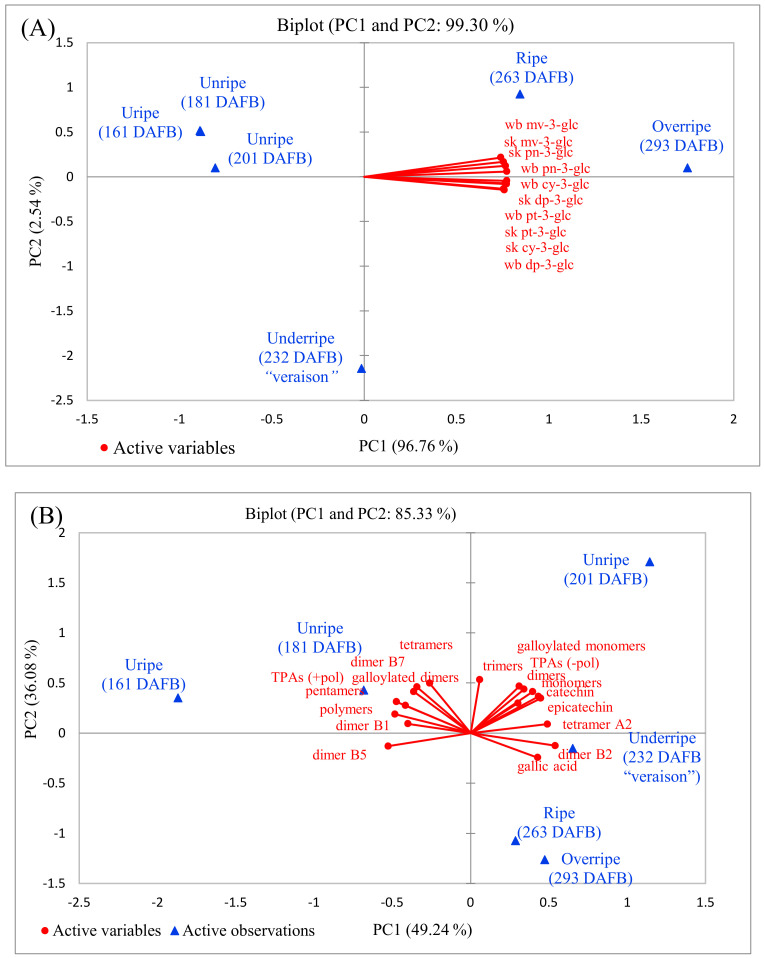
Biplot (PC1 × PC2) of scores and loadings for the PCA of all data of anthocyanins in skin and whole berry (**A**) and proanthocyanidins in seeds (**B**) of black ‘Isabel’ grape during six stages of development. Abbreviations: DAFB; days after full bloom, TPAs (+pol); Total proanthocyanidin with polymers, TPAs (−pol); Total proanthocyanidins without polymers.

**Figure 4 molecules-25-03845-f004:**
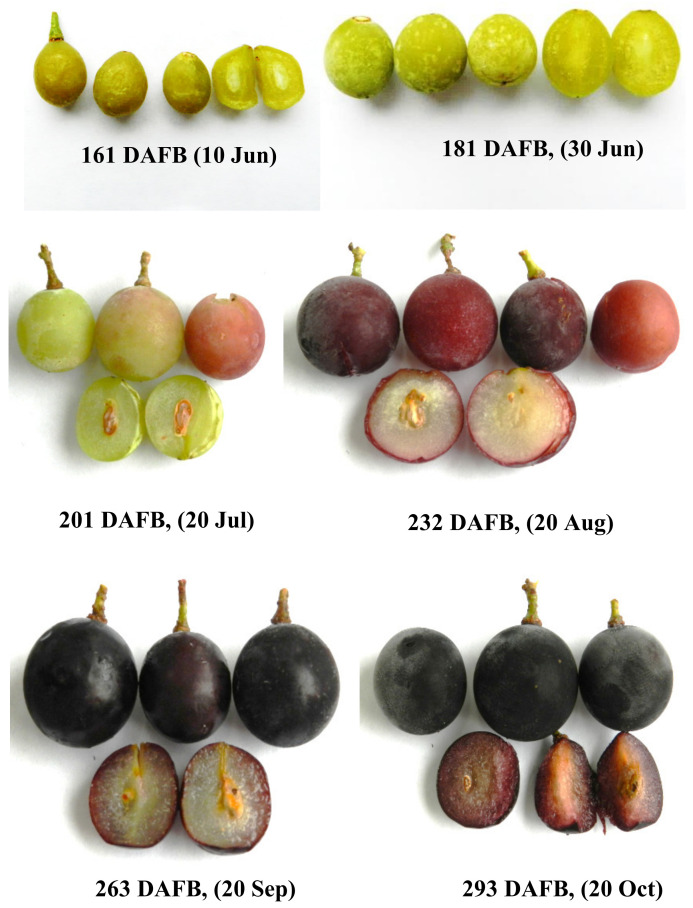
Berries of black ‘Isabel’ grape during six development stages. Scale bar = 0.5 cm.

**Table 1 molecules-25-03845-t001:** Phenolic compounds contents (TPC, TF and ACY) and antioxidant capacity values (DPPH and CUPRAC) in berries of black ‘Isabel’ grape during berry development.

DAFB ^Φ^	Berry Development	TPC ^χ^	TF ^δ^	ACY ^φ^	DPPH ^β^	CUPRAC ^β^
	**Whole berry**					
**161**	Unripe (green)	115.3 ± 9.4 ^f^	12.1 ± 0.1 ^d^	0.03 ± 0.05 ^f^	0.32 ± 0.07 ^f^	8.5 ± 0.8 ^e^
**181**	Unripe (green)	119.1 ± 4.1 ^e^	13.3 ± 3.2 ^d^	0.12 ± 0.01 ^e^	0.72 ± 0.28 ^e^	13.6 ± 0.5 ^d^
**201**	Unripe (green)	128.2 ± 6.3 ^d^	18.4 ± 0.1 ^d^	2.8 ± 0.2 ^d^	0.98 ± 0.06 ^d^	19.3 ± 0.4 ^c^
**232**	Underripe (reddish)	235.5 ± 1.0 ^c^	23.1 ± 0.3 ^c^	12.1 ± 0.1 ^c^	5.87 ± 0.03 ^c^	21.5 ± 0.2 ^b^
**263**	Ripe (red)	312.1 ± 3.3 ^b^	27.2 ± 0.1 ^b^	31.7 ± 2 ^b^	13.5 ± 0.1 ^b^	26.3 ± 1.3 ^a,b^
**293**	Overripe (dark blue black)	334.4 ± 8.1 ^a^	32.6 ± 1.3 ^a^	42.3 ± 2 ^a^	16.4 ± 0.3 ^a^	33.5 ± 10.2 ^a^
***R^2^*** **(*r*)**	TPC → DPPH	0.974 (0.987**)		TPC → CUPRAC	0.848 (0.921**)	
	TF → DPPH	0.924 (0.961**)		TF → CUPRAC	0.968 (0.984**)	
	ACY →DPPH	0.991 (0.976**)		ACY → CUPRAC	0.927 (0.912**)	
	**Skin**					
**161**	Unripe (green)	203.1 ± 6.3 ^d^	10.2 ± 0.3 ^e^	0.03 ± 0.05 ^d^	0.75 ± 0.07 ^f^	16.3 ± 1.1 ^f^
**181**	Unripe (green)	212.7 ± 9.1 ^c^	13.1 ± 0.1 ^d^	0.12 ± 0.05 ^d^	1.42 ± 0.08 ^e^	21.7 ± 1.3 ^e^
**201**	Unripe (green)	218.3 ± 19.4 ^c^	18.6 ± 0.1 ^c^	4.3 ± 0.4 ^c^	2.37 ± 0.34 ^d^	26.2 ± 0.4 ^d^
**232**	Underripe (reddish)	229.4 ± 12.7 ^b^	49.4 ± 1.2 ^b^	84.2 ± 3.2 ^b^	11.1 ± 0.2 ^c^	27.3 ± 0.4 ^c^
**263**	Ripe (red)	383.8 ± 15.6 ^b^	84.1 ± 1.3 ^b^	169.5 ± 4.4 ^a^	18.4 ± 0.2 ^b^	40.1 ± 2.4 ^b^
**293**	Overripe (dark blue black)	474.2 ± 7.2 ^a^	93.5 ± 4.0 ^a^	171.2 ± 2.0 ^a^	26.2 ± 0.3 ^a^	44.6 ± 2.1 ^a^
***R^2^*** **(*r*)**	TPC → DPPH	0.951 (0.972**)		TPC → CUPRAC	0.891 (0.944**)	
	TF → DPPH	0.979 (0.983**)		TF → CUPRAC	0.915 (0.956**)	
	ACY → DPPH	0.953 (0.968**)		ACY → CUPRAC	0.876 (0.936**)	
	**Seed**					
**201**	Unripe (green)	218.6 ± 7.2 ^b^	93.3 ± 2.2 ^b^	n.d.	74.2 ± 3.1 ^d^	189 ± 8.3 ^d^
**232**	Underripe (reddish)	259.1 ± 3.1 ^b^	100.6 ± 6.1 ^b^	n.d.	92.3 ± 1.4 ^c^	256 ± 3.5 ^c^
**263**	Ripe (red)	383.5 ± 1.3 ^a^	117.4 ± 2.4 ^a^	n.d.	95.7 ± 2.2 ^b^	296 ± 7.2 ^b^
**293**	Overripe (dark blue black)	384.2 ± 4.2 ^a^	118.5 ± 9.0 ^a^	n.d.	128.7 ± 4.6 ^a^	364 ± 11.6 ^a^
***R^2^ (r)***	TPC → DPPH	0.840 (0.917)		TPC → CUPRAC	0.934 (0.967*)	
	TF → DPPH	0.624 (0.790)		TF → CUPRAC	0.842 (0.917)	

(*n* = 3). Mean ± SD. Analysis of variance (one-way ANOVA) was used for comparisons. The mean values in columns followed by different letters at superscript indicate significant differences at *P* < 0.05 among berry development stages for whole berry, skin and seed, separately. *significant at *P* < 0.05. **significant at *P* < 0.01. ^χ^mg GAE/100 g fw, ^δ^mg QE/100 g fw, ^φ^mg cy-3-glu /100 g fw, ^β^µmol TE /g fw. Abbreviations: TPC; total phenolic compounds, TF; total flavonoid, ACY; total anthocyanin, DPPH; 2,2-diphenyl-1-picrylhydrazyl, CUPRAC; CUPric Reducing Antioxidant Capacity, ^Φ^DAFB; Days After Full Bloom.

**Table 2 molecules-25-03845-t002:** Changes in anthocyanin contents in skin and whole berry and proanthocyanidins contents in seed in ‘Isabel’grape during berry development.

Compound ^A^	Unripe (green) 161 DAFB	Unripe (green) 181 DAFB	Unripe (green) 201 DAFB	Underripe (reddish) 232 DAFB (veraison)	Ripe (red) 263 DAFB	Overripe (dark blue-black) 293 DAFB
**Anthocyanin (mg/kg fw)**						
Skin						
dp-3-glc	0.12 ± 0.01 ^a,^*	0.21 ± 0.01 ^a^	1.37 ± 0.03 ^b^	14.3 ± 0.3 ^c^	20.5 ± 1.1 ^d^	34.5 ± 11.3 ^e^
cy-3-glc	0.31 ± 0.01 ^a^	0.43 ± 0.02 ^a^	5.4 ± 0.1 ^b^	32.4 ± 0.4 ^c^	40.3 ± 1.4 ^d^	58.1 ± 2.4 ^e^
pt-3-glc	0.21 ± 0.01 ^a^	0.23 ± 0.01 ^a^	2.34 ± 0.1 ^a^	18.1 ± 0.3 ^b^	24.7 ± 1.3 ^c^	31.4 ± 1.2 ^d^
pn-3-glc	0.98 ± 0.01 ^a^	1.03 ± 0.07 ^a^	1.98 ± 0.02 ^a^	154.6 ± 2.3 ^b^	596.3 ± 3.2 ^c^	799.2 ± 8.1 ^d^
mv-3-glc	1.05 ± 0.04 ^a^	2.01 ± 0.06 ^a^	3.4 ± 0.2 ^a^	143.2 ± 2.2 ^b^	1499.2 ± 10.4 ^c^	1729.3 ± 8.5 ^d^
Whole berry						
dp-3-glc	ND	0.02 ± 0.0 ^a^	0.57 ± 0.08 ^b^	4.2 ± 0.2 ^c^	6.2 ± 0.1 ^d^	10.2 ± 0.4 ^e^
cy-3-glc	0.06 ± 0.01 ^a^	0.09 ± 0.01 ^a^	1.24 ± 0.12 ^b^	9.3 ± 0.2 ^c^	15.2 ± 0.4 ^d^	26.1 ± 1.2 ^e^
pt-3-glc	0.04 ± 0.0 ^a^	0.05 ± 0.01 ^a^	0.12 ± 0.01 ^a^	5.3 ± 0.2 ^b^	7.3 ± 0.2 ^c^	15.4 ± 1.1 ^d^
pn-3-glc	0.15 ± 0.01 ^a^	0.34 ± 0.03 ^a^	0.55 ± 0.06 ^a^	50.2 ± 1.4 ^b^	132.2 ± 1.6 ^c^	202.7 ± 3.3 ^d^
mv-3-glc	0.23 ± 0.03 ^a^	0.37 ± 0.05 ^a^	1.27 ± 0.06 ^a^	73.4 ± 0.1 ^b^	560.5 ± 14.2 ^c^	895.2 ± 14.5 ^d^
**Proanthocyanidins (PAs, %, dw)**						
Seed						
Monomers	1.47 ± 0.05 ^a^	3.1 ± 0.1 ^b,^*	13 ± 0.1 ^e^	6.7 ± 0.1 ^d^	4.39 ± 0.01 ^c^	3.06 ± 0.03 ^b^
Galloylated monomers	1.37 ± 0.04 ^b^	2.54 ± 0.03 ^e^	5.26 ± 0.03 ^f^	2.2 ± 0.1 ^d^	1.95 ± 0.04 ^c^	1.28 ± 0.02 ^a^
Dimers	1.43 ± 0.02 ^c^	2.15 ± 0.06 ^d^	3.01 ± 0.04 ^f^	2.5 ± 0.1 ^e^	1.61 ± 0.03 ^b^	1.44 ± 0.06 ^a^
Galloylated dimers	1.89 ± 0.02 ^f^	1.56 ± 0.02 ^e^	0.83 ± 0.03 ^d^	0.35 ± 0.01 ^c^	0.20 ± 0.01 ^a^	0.27 ± 0.02 ^b^
Trimers	0.67 ± 0.08 ^b^	0.85 ± 0.01 ^c^	0.92 ± 0.04 ^d^	0.72 ± 0.02 ^b^	0.50 ± 0.01 ^a^	0.55 ±0.02 ^a^
Tetramers	0.82 ± 0.03 ^d^	0.80 ± 0.01 ^c^	0.79 ± 0.03 ^c^	0.70 ± 0.01 ^b^	0.58 ± 0.02 ^a^	0.54 ±0.09 ^a^
Pentamers	0.73 ± 0.07 ^d^	0.63 ± 0.02 ^c^	0.64 ± 0.01 ^c^	0.52 ± 0.03 ^b^	0.45 ± 0.01 ^a^	0.41 ± 0.02 ^a^
Polymers	83 ± 1 ^e^	40 ± 0.1 ^d^	35 ± 0.2 ^b^	37 ± 1 ^c^	35 ± 0.4 ^b^	19 ± 0.2 ^a^
Gallic acid	ND	ND	0.29 ± 0.00 ^a^	0.63 ± 0.01 ^d^	0.52 ± 0.00 ^c^	0.35 ± 0.00 ^b^
Dimer B1	0.1 ± 0.00 ^d^	0.01 ± 0.00 ^a^	0.03 ± 0.00 ^c^	0.03 ± 0.00 ^c^	0.03 ± 0.00 ^c^	0.02 ± 0.00 ^b^
Catechin	0.58 ± 0.02 ^a^	2.02 ± 0.03 ^d^	5.41 ± 0.03 ^f^	3.01 ± 0.02 ^e^	1.64 ± 0.01 ^c^	0.95 ± 0.02 ^b^
Dimer B2	ND	0.02 ± 0.00 ^a^	0.08 ± 0.04 ^b^	0.10 ± 0.01 ^b^	0.07± 0.02 ^b^	0.08 ± 0.00 ^b^
Epicatechin	1.1 ± 0.1 ^a^	2.48 ± 0.04 ^c^	8.0 ± 0.1 ^f^	4.56 ± 0.03 ^e^	3.32 ± 0.02 ^d^	2.21 ± 0.12 ^b^
Tetramer A2	ND	ND	0.77 ± 0.06 ^c^	0.76 ± 0.02 ^c^	0.17 ± 0.00 ^a^	0.56 ± 0.05 ^b^
Dimer B7	2.18 ± 0.04 ^f^	2.33 ± 0.05 ^e^	1.72 ± 0.07 ^d^	1.23 ± 0.04 ^c^	0.89 ± 0.02 ^b^	0.76 ± 0.04 ^a^
Dimer B5	2.98 ± 0.1 ^e^	2.00 ± 0.22 ^d^	0 ± 0 ^a^	1.64 ± 0.13 ^c^	1.16 ± 0.29 ^b^	1.29 ± 0.01 ^b^
TPAs (+polymers)	95.5	49.5	59.5	48.7	44.7	36.6
TPAs (−polymers)	12.5	9.7	24.6	11.5	9.8	17.3

(*n* = 3). Mean ± SD. Analysis of variance (one-way ANOVA) was used for comparisons. The mean values in rows followed by different letters at superscript indicate significant differences at *P* < 0.05 among berry development stages for whole berry, skin and seed, separately. *Values are different from 0 with a significance level α = 0.05. ^A^Nomenclature abbreviations: dp-3-glc; delphinidin-3-*O*-glucoside, cy-3-glc; cyanidin-3-*O*-glucoside, pt-3-glc; petundin-3-*O*-glucoside, pn-3-glc; peonidin-3-*O*-glucoside, mv-3-glc; malvidin-3-*O*-glucoside, ND; not detected, TPAs (+polymers); TPAs with polymers, TPAs (−polymers); TPAs without polymers. Abbreviations: DAFB; Days After Full Bloom, fw; fresh weight, dw; dry weight.

**Table 3 molecules-25-03845-t003:** Matrix of correlation coefficients (*r*) of variables based on anthocyanin content (mg/kg fw) in the skin and whole berry in black ‘Isabel’ grape during berry development.

Variables	sk dp-3-glc	sk cy-3-glc	sk pt-3-glc	sk pn-3-glc	sk mv-3-glc	wb dp-3-glc	wb cy-3-glc	wb pt-3-glc	wb pn-3-glc	wb mv-3-glc
sk dp-3-glc^A^	-	**0.990 ***	**0.981**	**0.963**	**0.916**	**0.999**	**0.999**	**0.995**	**0.984**	**0.948**
sk cy-3-glc	-	-	**0.997**	**0.940**	**0.890**	**0.992**	**0.982**	**0.973**	**0.960**	**0.907**
sk pt-3-glc	-	-	-	**0.940**	**0.896**	**0.985**	**0.972**	**0.957**	**0.955**	**0.900**
sk pn-3-glc	-	-	-	-	**0.990**	**0.966**	**0.971**	**0.952**	**0.995**	**0.991**
sk mv-3-glc	-	-	-	-	-	**0.921**	**0.929**	**0.901**	**0.971**	**0.982**
wb dp-3-glc	-	-	-	-	-	-	**0.998**	**0.991**	**0.984**	**0.948**
wb cy-3-glc	-	-	-	-	-	-	-	**0.996**	**0.990**	**0.962**
wb pt-3-glc	-	-	-	-	-	-	-	-	**0.978**	**0.950**
wb pn-3-glc	-	-	-	-	-	-	-	-	-	**0.988**
wb mv-3-glc	-	-	-	-	-	-	-	-	-	-

^A^Nomenclature abbreviations: dp-3-glc; delphinidin-3-*O*-glucoside, cy-3-glc; cyanidin-3-*O*-glucoside, pt-3-glc; petundin-3-*O*-glucoside, pn-3-glc; peonidin-3-*O*-glucoside, mv-3-glc; malvidin-3-*O*-glucoside. *Values in bold are different from 0 with a significance level α = 0.05, sk; skin, wb; whole berry.

**Table 4 molecules-25-03845-t004:** Matrix of dominant rotated factor (F) loadings on anthocyanin concentration (mg/kg fw) in the skin and whole berry in black ‘Isabel’ grape during berry development.

Variables	F1	F2	F3	F4	F5	Observations	F1	F2	F3	F4	F5
sk dp-3-glc ^A^	**0.987 ***	0.011	0.002	0.000	0.000	Uripe (161 DAFB)	**0.989**	0.009	0.002	0.000	0.000
sk cy-3-glc	**0.959**	0.035	0.005	0.000	0.000	Unripe (181 DAFB)	**0.990**	0.008	0.002	0.000	0.000
sk pt-3-glc	**0.950**	0.030	0.020	0.000	0.000	Unripe (201 DAFB)	**0.999**	0.000	0.000	0.001	0.000
sk pn-3-glc	**0.974**	0.025	0.002	0.000	0.000	Underripe (232 DAFB)	0.002	**0.983**	0.015	0.000	0.000
sk mv-3-glc	**0.912**	0.078	0.010	0.000	0.000	Ripe (263 DAFB)	**0.939**	0.030	0.032	0.000	0.000
wb dp-3-glc	**0.990**	0.010	0.000	0.000	0.000	Overripe (293 DAFB)	**0.996**	0.000	0.004	0.000	0.000
wb cy-3-glc	**0.993**	0.003	0.004	0.000	0.000	Variability (%)	**96.76**	2.54	0.69	0.01	0.00
wb pt-3-glc	**0.972**	0.009	0.019	0.000	0.000	Cumulative (%)	**96.76**	99.30	99.99	100.00	100.00
wb pn-3-glc	**0.994**	0.006	0.000	0.000	0.000						
wb mv-3-glc	**0.947**	0.046	0.006	0.000	0.000						
Variability (%)	**96.76**	2.54	0.69	0.01	0.00						
Cumulative (%)	**96.76**	99.30	99.99	100.00	100.00						

^A^Nomenclature abbreviations: dp-3-glc; delphinidin-3-*O*-glucoside, cy-3-glc; cyanidin-3-*O*-glucoside, pt-3-glc; petundin-3-*O*-glucoside, pn-3-glc; peonidin-3-*O*-glucoside, mv-3-glc; malvidin-3-*O*-glucoside. *Values in bold are different from 0 with a significance level α = 0.05, DAFB: Days after full bloom sk; skin, wb; whole berry.

**Table 5 molecules-25-03845-t005:** Matrix of correlation coefficients (*r*) of variables based on proanthocyanidins content (%, fresh weight) in the seed in black ‘Isabel’ grape during berry development.

Variables	M	Gall M	D	Gall D	Tri	Tet	Pent	Pol	GaA	DB1	C	DB2	EC	TetA2	DB7	DB5	TPAs(+pol)	TPAs (−pol)
M	-	**0.927 ***	**0.902**	−0.301	0.593	0.192	0.074	−0.346	0.347	−0.302	**0.977**	0.581	**0.997**	0.742	-0.065	**−0.845**	−0.153	0.745
Gall M	-	-	**0.902**	0.002	0.777	0.431	0.318	−0.205	0.009	−0.277	**0.954**	0.274	**0.921**	0.477	0.247	−0.745	−0.021	0.705
D	-	-	-	−0.074	**0.821**	0.447	0.259	−0.250	0.190	−0.360	**0.969**	0.421	**0.908**	0.621	0.221	−0.633	−0.114	0.532
Gall D	-	-	-	-	0.470	**0.842**	**0.906**	0.793	**−0.918**	0.558	−0.188	**−0.916**	−0.338	−0.634	**0.945**	0.618	0.781	−0.132
Tri	-	-	-	-	-	0.802	0.657	0.095	−0.388	−0.139	0.723	−0.124	0.578	0.210	0.699	−0.244	0.213	0.426
Tet	-	-	-	-	-	-	**0.965**	0.664	−0.646	0.405	0.323	−0.583	0.163	−0.238	**0.943**	0.281	0.708	0.090
Pent	-	-	-	-	-	-	-	0.801	−0.731	0.599	0.169	−0.705	0.033	−0.352	**0.929**	0.377	**0.854**	0.099
Pol	-	-	-	-	-	-	-	-	−0.567	**0.913**	−0.314	−0.768	−0.386	−0.563	0.662	0.720	**0.963**	−0.256
GaA	-	-	-	-	-	-	-	-	-	−0.369	0.257	**0.904**	0.386	0.658	**−0.829**	−0.480	−0.582	0.007
DB1	-	-	-	-	-	-	-	-	-	-	−0.355	−0.563	−0.356	−0.355	0.355	0.602	**0.928**	−0.066
C	-	-	-	-	-	-	-	-	-	-	-	0.496	**0.980**	0.668	0.085	−0.777	−0.146	0.655
DB2	-	-	-	-	-	-	-	-	-	-	-	-	0.608	**0.876**	−0.779	−0.727	−0.694	0.353
EC	-	-	-	-	-	-	-	-	-	-	-	-	-	0.735	−0.089	**−0.858**	−0.208	0.699
TetA2	-	-	-	-	-	-	-	-	-	-	-	-	-	-	−0.495	−0.697	−0.398	0.652
DB7	-	-	-	-	-	-	-	-	-	-	-	-	-	-	-	0.433	0.659	−0.078
DB5	-	-	-	-	-	-	-	-	-	-	-	-	-	-	-	-	0.553	−0.697
TPAs (+pol)	-	-	-	-	-	-	-	-	-	-	-	-	-	-	-	-	-	0.013
TPAs (-pol)	-	-	-	-	-	-	-	-	-	-	-	-	-	-	-	-	-	-

*Values in bold are different from 0 with a significance level α = 0.05. Abrreviations: M; Monomers. Gall M; Galloylated monomers. D. Dimers. Gall D; Galloylated dimers. Tr; Trimers. Tet; Tetramers. Pent; Pentamers. Pol; Polymers. GaA; Gallic acid. DB1; Dimer B1. C; Catechin. DB2; Dimer B2. EC; Epicatechin. TetA2; Tetramer A2. DB7; Dimer B7. DB5; Dimer B5. TPAs (+pol); Total proanthocyanidin with polymers, TPAs (−pol); Total proanthocyanidins without polymers.

**Table 6 molecules-25-03845-t006:** Matrix of dominant rotated factor (F) loadings on proanthocyanidins (%, fresh weight) in the seed in black ‘Isabel’ grape during berry development.

Variables	F1	F2	F3	F4	F5	Observations	F1	F2	F3	F4	F5
Monomers	**0.560 ***	0.402	0.026	0.000	0.012	Uripe (161 DAFB)	**0.917**	0.024	0.058	0.001	0.000
Galloylated monomers	0.288	**0.658**	0.004	0.009	0.041	Unripe (181 DAFB)	0.354	0.103	**0.540**	0.003	0.000
Dimers	0.347	**0.575**	0.009	0.067	0.003	Unripe (201 DAFB)	0.371	0.607	0.007	0.013	0.001
Galloylated dimers	**0.673**	0.295	0.021	0.006	0.005	Underripe (232 DAFB)	**0.518**	0.021	0.042	0.369	0.050
Trimers	0.010	**0.855**	0.093	0.007	0.035	Ripe (263 DAFB)	0.074	**0.770**	0.003	0.004	0.149
Tetramers	0.207	**0.748**	0.004	0.037	0.004	Overripe (293 DAFB)	0.138	**0.723**	0.000	0.104	0.035
Pentamers	0.354	**0.637**	0.007	0.002	0.000	Variability (%)	49.24	**36.08**	8.15	4.35	2.18
Polymers	**0.705**	0.108	0.156	0.022	0.008	Cumulative (%)	49.24	**85.33**	93.48	97.82	100.00
Gallic acid	**0.547**	0.177	0.099	0.172	0.004						
Dimer B1	0.483	0.027	**0.489**	0.001	0.001						
Catechin	0.469	**0.512**	0.000	0.012	0.007						
Dimer B2	**0.874**	0.046	0.034	0.030	0.015						
Epicatechin	**0.600**	0.364	0.013	0.003	0.019						
Tetramer A2	**0.718**	0.024	0.100	0.003	0.154						
Dimer B7	0.398	**0.515**	0.080	0.006	0.001						
Dimer B5	**0.832**	0.051	0.001	0.066	0.051						
TPAs (+pol)	**0.523**	0.231	0.245	0.000	0.002						
TPAs (−pol)	0.274	0.269	0.086	**0.341**	0.031						
Variability (%)	49.24	36.08	8.15	**4.35**	2.18						
Cumulative (%)	49.24	85.33	93.47	**97.82**	100.00						

*Values in bold are different from 0 with a significance level α = 0.05. Abbreviations: DAFB; Days after full bloom, TPAs (+pol); Total proanthocyanidin with polymers, TPAs (−pol); Total proanthocyanidins without polymers.

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
