# Peer review of "Accumulation of Phenolic Compounds and Antioxidant Capacity during Berry Development in Black ‘Isabel’ Grape (Vitis vinifera L. x Vitis labrusca L.)"

_molecules, 2020, doi:10.3390/molecules25173845_

Round 1
Reviewer 1 Report
This work was well conducted and, in my opinion, has merit for publication. However, I make some considerations below:
Line 27: The present study characterized phenolic compounds in the Isabel grape for the first time in the region studied, since there are several studies on the characterization of the Isabel grape in other world regions.
Line 54: “Concord”Lines 355-358: Considering that it is a hybrid V. labrusca grape, why did they not use external patterns of diglucoside anthocyanins?
Lines 125-127: How do you explain why a hybrid V. labrusca grape does not have diglucoside anthocyanins?
Table 2: How the trimers, tetramers and others were quantified. These compounds are not included in the list of external standards. If it was a NIST library, it cannot be considered quantification, just an attempt at identification.
Author Response
Manuscript ID: molecules-897371
Reviewer 1
This work was well conducted and, in my opinion, has merit for publication. However, I make some considerations below:
Line 27: The present study characterized phenolic compounds in the Isabel grape for the first time in the region studied, since there are several studies on the characterization of the Isabel grape in other world regions.
Respond: 28-31…The aim of this study was to investigate phenolic compounds and antioxidant capacity in the grape during berry development, that has been not reported yet from the region. A trend to an increase in antioxidant capacity values and total phenolic compounds, flavonoid and anthocyanin contents were observed from un-ripened to overly-ripened berries. In addition
Line 54: “Concord”Lines 355-358: Considering that it is a hybrid V. labrusca grape, why did they not use external patterns of diglucoside anthocyanins?
Respond:
According to the statement of Brauch et al., at pH 3.6, half-lives of 3-O-glycosides are substantially shorter than those of respective 3,5-O-diglycosides, at pH 2.2, an inverse stability behavior was observed. To achieve best resolution of picks, we used a mixture of acetonitrile and acidified water by formic acid till pH=1,9. The monoglycoside anthocyanins are more dominant and more stable in this solution of mobile phase.
Reference: Studies into the Stability of 3-O-Glycosylated and 3,5-O-Diglycosylated Anthocyanins in Differently Purified Liquid and Dried Maqui (Aristotelia chilensis (Mol.) Stuntz) Preparations during Storage and Thermal Treatment, Brauch J.E; Kroner, M; Ralf M. Schweiggert, R.M; Carle, R. J. Agric. Food Chem. 2015, 63, 39, 8705–8714
Lines 125-127: How do you explain why a hybrid V. labrusca grape does not have diglucoside anthocyanins?
Respond:
Monoglycoside anthocyanins were used for the identification and quantitation since they were the main compounds identified in the anthocyanin profiles of ‘Isabel’ grapes.
Are you sure that the diglucoside ones were not identified in grape samples? In that case, you could indicate that diglycoside anthocyanins could be found in Isabel grapes because its a hybrid grape cultivar, but they were not identified in this work because they can be found at low concentration as reported by Yamamoto et al. 2015.
It has been known that Vitis vinifera grapes have anthocyanidin-3-O-monoglucoside as the major anthocyanins, whereas diglucoside conjugated anthocyanins are mainly present in other grapes, such as V. amurensis, V. riparia, V. rupestris, and their hybrids. Furthermore, the color of some hybrids such Vitis vinifera x Vitis labrusca is dependent manly of monoglycosides since diglycoside are more susceptible for browning. By considering these explanations, necessary additions were in the text as follows;
Lines 158-162
It has been known that V. vinifera grapes composed of anthocyanidin-3-O-monoglycosides as major anthocyanins, whereas diglycosides are mainly present in other grapes, such as V. amurensis, V. riparia, V. rupestris, and their hybrids. Furthermore, the color of some hybrids such V. vinifera x V. labrusca is dependent mainly of monoglycosides since diglycosides are more susceptible for browning. [19,20].
and lines 197-204
“As indicated above ‘Isabe’l is a hybrid grape cultivar (V. vinifera × V. labrusca) and the occurrence of anthocyanidin 3,5-diglucosides is expected, in line with previous findings [9] and other hybrid grape cultivars [2, 27,29]. Due to that, the quantification of these compounds is made using the calibration curve of the most similar compound: malvidin 3-glucoside is used for anthocyanidin 3-glucosides, and malvidin 3,5-diglucoside for anthocyanidin 3,5-diglucosides. However, a previous study [10] showed the anthocyanin monoglucosides were the main ones in the anthocyanin profiles of ‘Isabel’ grape juice, with a lower proportion of diglucoside anthocyanins”
Table 2: How the trimers, tetramers and others were quantified. These compounds are not included in the list of external standards. If it was a NIST library, it cannot be considered quantification, just an attempt at identification.
Respond:
Please see line 481-484: “The calibration curve of the trimer procyanidin C1 was in the range of 8.9 to 298 mg/L and was used for the trimeric proanthocyanidins. The calibration curve of cinnamtannin A2 was in the range of 8.1 to 271.6 mg/L and was used for the quantification of tetrameric proanthocyanidins and as an equivalent for pentameric and polymeric proanthocyanidins.”
best wishes
Corresponding author

Reviewer 2 Report
The introduction could be improved, better argue the importance of knowledge of composition and its influence to economic aspect. Improve also the discussion (and references) about the implication of antioxidant activity.
Pg 3. Argue about the significance of similarity and differences. Argue about the conseguences of differences in term of taste, health benefits and economic implications.
Pg 9. What is the importance that results are in agreement with those purchesed by other researchers (ex. North Arerica etc)
Which of the factorsi influencing the anthocyanin (24) content could be consiedered important in this study?
All the article is very good but in general it could be better discussed.
Author Response
Reviewer 2
The introduction could be improved, better argue the importance of knowledge of composition and its influence to economic aspect. Improve also the discussion (and references) about the implication of antioxidant activity.
Respond: It was addressed in different paragraphs in the “Introduction” section;
- i) economic and health importance: 46-53 and 67-68 in part
Grapes (Vitis spp L.) are one of the most important horticultural crops across the world and have widely received considerable attention because of their polyphenols [1-3]. Polyphenols are also a major source of phytochemicals in grape, mainly including stilbenes (resveratrol), flavonols, flavanols, procanthoyanidins, anthocyanins and phenolic acids in varying different amounts in skin, flesh and seed in grape [3]. They show a wide spectrum of health promoting properties including antioxidant activity, anti-inflammatory, anticarcinogenic and antibacterial activities [1-4]. The annual world grape production for 2017 exceeded 77 million tons. Turkey is one of the world’s three largest grape producers, at 9 million tons [5].
and line 68
……Also, it has a market value and is therefore widely sold as a table grape in coastal cities in the region.
Pg 9. What is the importance that results are in agreement with those purchesed by other researchers (ex. North Arerica etc).
The current results are reported for the first time from the Black Sea region for ‘Isabel’ grape with the present study. As stated in the discussion section, although studies have been reported for ‘Isabel’ grape phenolics from America, European and Asian countries, the present results from the Black Sea region are comparable. And it was discussed entirely in the section.
Which of the factors influencing the anthocyanin (24) content could be considered important in this study?
Furthermore, Shoeva et. al [27] demonstrated that even in universal metabolic pathways, some species-specific regulatory features of structural genes are present. Their results indicated in the anthocyanin biosynthesis pathway (ABP), genes may be regulated by ABP-specific regulatory factors, and their expression levels may be strongly associated with anthocyanin pigmentation, or they may be expressed independently of pigmentation. It has been known that ‘Isabella’ grape varieties are resistant to fungal diseases and they can be grown in regions with cool and humid climates [28]. ‘Isabel’ (or Isabella) grape varieties used for grape cultivation in Black Sea region are not affected by fungal diseases, there is no need for the use of drugs in the grape’s cultivation, but some nematodes can cause serious leaf and fruit losses, especially in black ‘Isabel’ grape (reference: personal communication with authorities in Republic of Turkey of Ministry of Agriculture and Forestry-Trabzon Directorate of Agricultural Organization).
best wishes
Corresponding author
All the article is very good but in general it could be better discussed.
